# Redundancy-Free Message Passing for Graph Neural Networks

**Rongqin Chen** [1,2]
chen.rongqin@connect.um.edu.mo

**Shenghui Zhang** [1]
zhang.shenghui@connect.um.edu.mo

**Leong Hou U** [1]
ryanlhu@um.edu.mo

**Ye Li** [2]
ye.li@siat.ac.cn

[1] State Key Laboratory of Internet of Things for Smart City, University of Macau, Macao SAR, China
[2] Shenzhen Institutes of Advanced Technology, Chinese Academy of Sciences, China

## Abstract

Graph Neural Networks (GNNs) resemble the Weisfeiler-Lehman (1-WL) test; they iteratively update the representation of each node by aggregating information from WL subtrees. However, despite the computational superiority of the iterative aggregation scheme, it introduces redundant message flows to encode nodes. We find that the redundancy in message passing prevented conventional GNNs from propagating the information of long-length paths and learning graph similarities. In order to address this issue, we proposed a Redundancy-Free Graph Neural Network (RFGNN), in which the information of each path (of limited length) in the original graph is propagated along a message flow devoid of duplicate edges. Our rigorous theoretical analysis demonstrates the following advantages of RFGNN: (1) The height $k$ RFGNN is strictly more powerful than the $k$-layer WL-test based GNNs; (2) RFGNN efficiently propagates structural information in original graphs, avoiding the over-squashing issue; and (3) RFGNN encodes graphs of small graph edit distances into similar representations in practice. The experimental evaluation of graph property prediction tasks validates our theoretical statements. And RFGNN achieves the best performance in most challenges.

## 1 Introduction

Graph Neural Networks (GNNs) generally imitate the Weisfeiler-Lehman (1-WL) test [1] in the sense that they iteratively update the representation of each node of a graph by aggregating information from its neighbors [2], which are referred to as WL-test based GNNs in this study. The iterative neighborhood aggregation scheme encodes trees (called WL subtrees [3]), where a node's WL subtree children are its graph neighbors, so that a WL subtree is a height-balanced tree. Research on GNNs has made great strides and achieved remarkable success in recent years. Nevertheless, the vast majority of studies do not break the paradigm of repetitive neighborhood aggregation, which provides computational superiority but generates negative consequences due to message flow redundancy [1] in tree-like encoding. Figure 1 illustrates an example of message flow redundancy in a computational graph. Message flow redundancy is prevalent, especially when the number of message passing iterations (a.k.a. neural network layers) increases.

---

[1] A recent study has been conducted on the computation of redundancy in GNNs, e.g. [4]. We add a note that the computational redundancy is orthogonal to the message passing redundancy.

36th Conference on Neural Information Processing Systems (NeurIPS 2022).

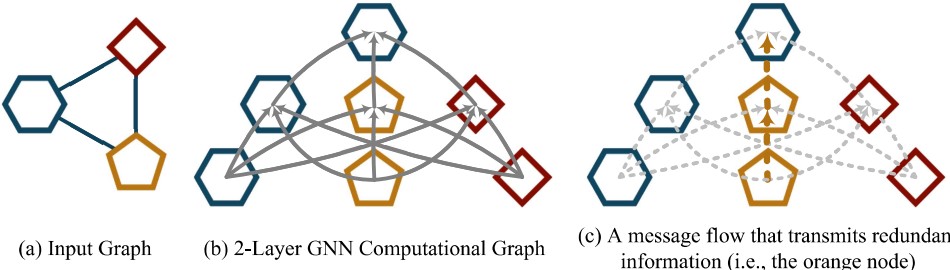

(a) Input Graph    (b) 2-Layer GNN Computational Graph    (c) A message flow that transmits redundant information (i.e., the orange node)

Figure 1: Redundancy exists in message flows, which are propagation chains in a GNN computational graph that deliver the information from input nodes to output nodes.

The redundancy phenomenon is accompanied by the problem of *over-squashing*, which refers to the phenomenon in which the information originating from distant nodes can be distorted, thus limiting the efficiency of message passing for tasks that rely on long-distance interactions [5]. This issue is attributable to the bottleneck of compressing a rapidly growing number of neighboring nodes into a vector of finite length [6] and to the failure of certain input features to influence the output representation [5]. However, in previous studies, the understanding of over-squashing and the explanation for why it occurs are still deficient. In this study, we carefully analyze the influence of an input feature on an output representation along a *trail* (extracted from the original graph), and we observe that longer trails typically have a smaller impact than shorter trails in the neighborhood aggregation process. One key reason behind this is that short trails usually result in a great number of redundant message flows in the computational graph.

According to previous research [7, 8, 9, 10], a fundamental problem of graph learning is to quantify the similarities between graphs. In the real world, many graphs, such as molecules, have structures at both coarse levels (such as chains, rings, and trees) and fine levels (like chemical bonds and functional groups). Some scholars [11] believe that an ideal method of graph similarity measurement should be able to capture subgraphs at different levels of granularity. However, 1-WL [3] can only capture the graph similarity at coarse levels because WL subtrees are height-balanced trees. Besides, WL-test based GNNs tend to encode two graphs that yield similar message flows as similar representations. However, even if only one edge in the two original graphs is different, it can lead to two very different message flow sets due to message flow redundancy. Thus, WL-test based GNNs may have insufficient expressive power to measure graph similarity.

We propose a Redundancy-Free Graph Neural Network (RFGNN) to address the aforementioned issues. We first eliminate message flow redundancy with a concept called *path-search-tree* (a.k.a. breadth-first tree), which is constructed by running a breadth-first search (BFS) from a node [12]. Since there are no repeated nodes in each path in the path-search-tree, by using the tree as the message passing graph, the resulting model would not suffer from message flow redundancy. However, this simple path-search-tree model cannot capture some graph structures, e.g., ring. To enhance the expressiveness of the model, we define a new concept called *extended path (epath)*, which can represent a ring structure when its starting and finishing nodes are the same. Based on epath, we propose a concept called *Truncated ePaths Tree (TPT)*, an epath-search-tree obtained by running a BFS, where all epaths of limited length starting from the node are accessed. Given a TPT (of a graph node), RFGNN is a neural network to perform bottom-up message passing. Finally, we collect the representations of all TPTs of a graph in order to update the representation of the entire graph.

**In comparison to prior research in the field of GNNs, the new findings of this work were summarized as follows**: 1) To the best of our knowledge, this is the first study that investigates the theoretical and practical implications of message flow redundancy in GNNs; 2) We proved that the height $k$ RFGNN is strictly more expressive than $k$-layer WL-test based GNNs; 3) De-redundancy balances the influence of epaths starting at the same node during message passing, propagates structural information in original graphs efficiently, and reduces over-squashing; 4) RFGNN always generates similar message flows for similar graphs compared to WL-test based GNNs; hence, RFGNN can learn better representations of similar graphs than WL-test based GNNs.

The rest of this paper is structured as follows. Section 2 outlines the basic theories of GNNs. In Section 3, we demonstrate that message flow redundancy is prevalent in GNNs and analyze why it contributes to the issue of over-squashing in WL-test based GNNs. The framework of RFGNN is presented in Section 4, which not only eliminates the redundancy in computational graphs but also provides several additional advantages. Related work is then discussed in Section 5. Section 6 empirically validates RFGNN on a variety of graph property prediction challenges and demonstrates that RFGNN achieves state-of-the-art performance on the vast majority of tasks. The conclusion is given in Section 7.

## 2   Preliminaries

We begin by reviewing the most commonly used message passing paradigm and its derivative GNNs, as well as introducing our notations. Let $G\left(V, E\right) \in \mathcal{G}$ be a graph, where $V = \mathcal{V}(G) = \{1, 2, \ldots\}$, $E = \mathcal{E}(G) \subseteq V \times V$, and $\mathcal{G}$ is the graph set consisting of all possible graphs. The nodes and edges in $G$ can have feature vectors associated with them, denoted by $\mathbf{x}_u$ for $u \in V$ and $\mathbf{e}_{u,v}$ for $(u, v) \in E$. In this work, we consider the graph property prediction problem as follows: given a set of graphs $\{G_1, \ldots, G_N\} \subseteq \mathcal{G}$ and their labels $\{y_1, \ldots, y_N\} \subseteq \mathcal{Y}$, we aim to learn a representation vector $\mathbf{h}_G$ that helps to predict the label of an entire graph by a predictor, $y_G = \rho(\mathbf{h}_G)$.

**Message passing and graph neural networks.**   GNNs iteratively update the representation of a node by aggregating its neighbor information. The $l$-th layer of a GNN can be described as

$$\mathbf{h}_u^{(l)} = \psi^{(l)}\left(\mathbf{h}_u^{(l-1)}, \cup_{v \in \mathcal{N}(u)} \phi^{(l)}\left(\mathbf{e}_{(u,v)}^{(l-1)}, \mathbf{h}_v^{(l-1)}\right)\right), \tag{1}$$

where $\psi^{(l)}$ and $\phi^{(l)}$ denote differentiable functions such as MLPs (Multi Layer Perceptrons), $\cup$ denotes a differentiable and permutation invariant aggregator, (e.g., *sum* and *mean*), $\mathbf{h}_u^{(l)}$ is the node $u$ representation, and $\mathbf{e}_{(u,v)}^{(l-1)}$ is the edge $(u, v)$ representation transformed from $\mathbf{e}_{(u,v)}$ by a differentiable function such as a linear function. We initialize $\mathbf{h}_u^{(0)} = \mathbf{x}_u$. The readout $\chi$ function aggregates node representations from the final layer to obtain the entire graph representation $\mathbf{h}_G$:

$$\mathbf{h}_G = \chi\left(\left\{\mathbf{h}_u^{(k)} \mid u \in G\right\}\right). \tag{2}$$

The readout $\chi$ can be a simple permutation invariant function such as *sum*.

**Weisfeiler-Lehman (WL) test and WL-subtree.**   The graph isomorphism problem is to identify whether two graphs are topologically equivalent, i.e., isomorphic. The Weisfeiler-Lehman (1-WL) test [1] iteratively updates the label of a node by hashing the labels of its own and neighbors. If the node labels differ in any iteration, then 1-WL considers the graph to be non-isomorphic. Based on 1-WL, Shervashidze et al. [3] propose WL-subtree, which is a kernel method to measures the similarity of graphs based on the number of distinct node labels, where a node label indicates a WL subtree. In consequence, the graph similarity is then measured by the number of distinct WL subtrees.

**Computational graph.**   Computational graphs represent neural networks as graphs on which gradients can be conveniently computed using chaining laws [13]. In message passing GNNs, the computational graph of a node is the message passing graph that aggregates neighborhood information into the node. The computational graph is jointly decided by the adopted message passing paradigm and the structure of the input graph.

## 3   Message Flow Redundancy and Practical Implications

In this section, we first explain the redundancy issue in computational graphs and then discuss the over-squashing issue. Without loss of generality, we discuss the computational graph of a $k$-layer WL-test based GNN for (the representation of) a node. All the proofs can be found in Appendix A.

### 3.1 Message Flow Redundancy in Computational Graph

**Definition 1** (Message flow). *A message flow $p$ in a computational graph depicts the propagation chain of an input feature from an input node to an output node, where $p(i)$ denotes the $i$-th node of $p$ (zero based index).*

A WL-test based GNN iteratively updates the node representation by aggregating the neighbor information, where the message passing process (of the neighbor information) is analogous to the traverse of the graph by walks, defined as follows.

**Definition 2** (Walk). *A walk is a sequence of graph nodes and edges formed by a graph traversal that permits backtracking and self-looping.*

**Corollary 1.** *A message flow for output node $u$ in the computational graph is a walk of the same length that ends at $u$ in the original graph.*

**Definition 3** (Trail). *A trail is a succession of edges with no repetition, where $tr(i)$ refers to the $i$-th node of $tr$ (zero based index).*

For instance, $u_1 \rightarrow u_2 \rightarrow u_3 \rightarrow u_1 \rightarrow u_4$ is a valid trail, which has no repeating edges.

**Lemma 1.** *The message flows that propagate the information of a trail are the walks of the trail from its beginning node to its ending node.*

**Message flow redundancy.** A trail can not be compressed since there is no repeated edge in the trail. As illustrated in Figure 2, we can transform a trail into message flows via unordered sampling with replacement, a sampling fashion in which repetition is allowed and ordering does not matter [14, Section 2.1.4]. Message flows derived from the same trail carry identical trail-related information. The backtracking and self-looping edges in message flows do not provide additional information and are therefore redundant.

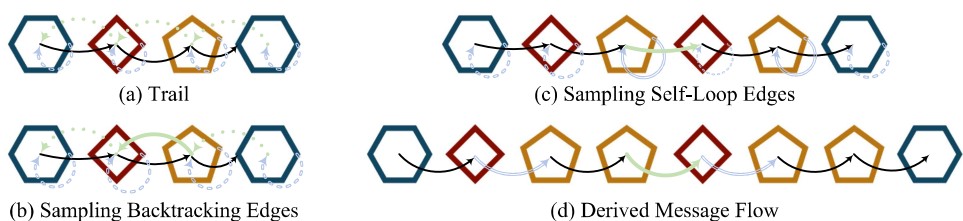

(a) Trail

(b) Sampling Backtracking Edges

(c) Sampling Self-Loop Edges

(d) Derived Message Flow

Figure 2: The process of deriving a message flow from a trail: Firstly, backtracking edges are sampled and placed in the trail. Then, self-loop edges are sampled and inserted into the trail.

**Message flow redundancy is prevalent.** Suppose there is a trail of length $r$ and a message flow of length $k$ derived from the trail. The backtracking edge of a self-loop edge is also a self-loop edge. Thus, we begin by sampling $s$ backtracking edges from the $r$ backtracking edges. After the backtracking edges are placed, we sample $k - r - 2s$ self-loop edges from the $r + 1 + 2s$ self-loop edges. Lemma 2 demonstrates that message flow redundancy is widespread because a trail shorter than the number of GNN layers would result in many message flows with backtracking and self-loop edges.

**Lemma 2.** *Given a length-$r$ trail, the number of length-$k$ message flows derived from the trail is*

$$\sum_{s=0}^{\lfloor \frac{k-r}{2} \rfloor} r^s (r + 1 + 2s)^{(k-r-2s)},$$

*where $1 \leq r \leq k$.*

### 3.2 Redundancy Causes Message Over-squashing

Message flow redundancy in computational graphs comes with a phenomenon known as *over-squashing*, which occurs when GNNs are unable to propagate information effectively across a graph

[6]. Recently, Topping et al. [5] have proposed the *Jacobian* of a GNN-output to quantify the effect of over-squashing.

**The influence of a change.** Assume there is a graph with nodes $u$ and $v$, as well as a $k$-layer WL-test based GNN to update node representations. The *Jacobian* measures how a change in the input feature $\mathbf{x}_v$ affects the node output $\mathbf{h}_u^{(k)}$ as

$$\partial \mathbf{h}_u^{(k)} / \partial \mathbf{x}_v. \tag{3}$$

Similar to [5], to avoid heavy notations, the node attributes and representations are assumed to be scalar; this assumption simplifies the analysis, whereas the vector case yields equivalent results.

**Lemma 3.** *Assume there is a graph with node set $V$, nodes $u, v \in V$, and the computational graph of a $k$-layer WL-test based GNN for $u$. If all length-$k$ paths in the computational graph for $u$ are activated with the same probability, we can measure the **relative influence of the input feature $\mathbf{x}_v$ on the node output $\mathbf{h}_u^{(k)}$**, on average, as*

$$\mathbb{E}\left(\frac{\partial \mathbf{h}_u^{(k)} / \partial \mathbf{x}_v}{\sum_{v' \in V} \partial \mathbf{h}_u^{(k)} / \partial \mathbf{x}_{v'}}\right) = \frac{\left[\prod_{l=k}^{1} \widetilde{A}\right]_{u,v}}{\sum_{v' \in V} \left[\prod_{l=k}^{1} \widetilde{A}\right]_{u,v'}}, \tag{4}$$

*where $\widetilde{A} = A + I$ is the adjacency matrix of the graph with self-loop.*

**The influence of a change along a message flow.** Assume there is a graph, the computational graph of a $k$-layer WL-test based GNN and its message flow $p$. The effect of a change in the node feature $\mathbf{x}_{p(0)}$ on the output node $\mathbf{h}_{p(k)}^{(k)}$ along the message flow $p$ (a.k.a **the influence of $\mathbf{x}_{p(0)}$ on $\mathbf{h}_{p(k)}^{(k)}$ along $p$**) can be computed as

$$\frac{\partial \mathbf{h}_{p(k)}^{(k)}}{\partial p} = \prod_{l=k}^{1} \frac{\partial \mathbf{h}_{p(l)}^{(l)}}{\partial \mathbf{h}_{p(l-1)}^{(l-1)}}. \tag{5}$$

**Lemma 4.** *Assume there is a graph with node set $V$, the computational graph of a $k$-layer WL-test based GNN for a node, and a message flow $p$ in the computational graph. We compute **the relative influence of $\mathbf{x}_{p(0)}$ on $\mathbf{h}_{p(k)}^{(k)}$ along $p$**, on average, as*

$$\mathbb{E}\left(\frac{\partial \mathbf{h}_{p(k)}^{(k)} / \partial p}{\sum_{p' \in P} \partial \mathbf{h}_{p(k)}^{(k)} / \partial p'}\right) = \frac{\prod_{l=k}^{1} \widetilde{A}_{p(l),p(l-1)}}{\sum_{v' \in V} \left[\prod_{l=k}^{1} \widetilde{A}\right]_{p(k),v'}}, $$

*where $\widetilde{A} = A + I$ is the adjacency matrix of the graph with self-loop and $P$ is the set of message flows that end at $tr(k)$.*

**The influence of a change along a trail.** Given a graph with a trail $tr$ and the computational graph of a $k$-layer WL-test based GNN for $tr$'s head node. Let $P^*$ be the set of message flows derived from $tr$ within the computational graph for the node $tr(k)$. How a change in $\mathbf{x}_{tr(0)}$ will be propagated along $tr$ and affect $\mathbf{h}_{tr(r)}^{(k)}$ (a.k.a **the influence of $\mathbf{x}_{tr(0)}$ on $\mathbf{h}_{tr(r)}^{(k)}$ along $tr$**), denoted as $\partial \mathbf{h}_{tr(r)}^{(k)} / \partial tr$, is computed as the summation of the influence of $\mathbf{x}_{tr(0)}$ on $\mathbf{h}_{tr(r)}^{(k)}$ along all message flows belonging to $P^*$, i.e.,

$$\frac{\partial \mathbf{h}_{tr(r)}^{(k)}}{\partial tr} = \sum_{p' \in P^*} \frac{\partial \mathbf{h}_{tr(r)}^{(k)}}{\partial p'}. \tag{6}$$

**Corollary 2.** *Assume there is a graph with a trail $tr$ and the computational graph of a $k$-layer WL-test based GNN for $tr$'s head node. We compute **the relative influence along $tr$**, on average, as*

$$\mathbb{E}\left(\frac{\partial \mathbf{h}_{tr(r)}^{(k)} / \partial tr}{\sum_{p' \in P} \partial \mathbf{h}_{p(k)}^{(k)} / \partial p'}\right) = \frac{\sum_{p' \in P^*} \prod_{r=k}^{1} \widetilde{A}_{p'(r),p'(r-1)}}{\sum_{v' \in V} \left[\prod_{r=k}^{1} \widetilde{A}\right]_{p(k),v'}}, $$

where $\widetilde{A} = A + I$ is the adjacency matrix of the graph with self-loop and $P^*$ is the set of message flows derived from $tr$.

The relative influence along a trail is proportional (in terms of probability) to the number of message flows derived from the trail, according to Corollary 2. According to Lemma 2, a trail typically produces fewer message flows than its child trail. Therefore, the effect along a lengthy path is always less than that of its sub-path. Thus, **redundancy in the message passing paradigm is a significant element in the occurrence of over-squashing**. Topping et al. [5] stated the phenomenon of over-squashing occurs when the absolute value of the *Jacobian* $\partial \mathbf{h}_u^{(k)} / \partial \mathbf{x}_v$ is small. Supplementally, we find the phenomenon of over-squashing would occur even if the absolute value of the *Jacobian* $\partial \mathbf{h}_u^{(k)} / \partial \mathbf{x}_v$ is large; keeping the influence along $tr$ constant, the relative influence along $tr$ decreases as the absolute value of the *Jacobian* $\partial \mathbf{h}_{tr(k)}^{(k)} / \partial \mathbf{x}_{tr(0)}$ increases.

# 4    Redundancy-Free Graph Neural Network

In the last section, we discussed message flow redundancy and its practical implications. To overcome this issue, we propose Redundancy-Free GNN (RFGNN), a kind of message passing GNN that is not only message flow redundancy-free but also improves the overall expressive power and message passing capability and thus generates better performance. All proofs for RFGNN can be found in Appendix B, and the space and time complexity analysis can be found in Appendix C.

## 4.1    The RFGNN Framework

We first discuss the redundancy-free message passing paradigm based on a concept called extended path, $epath$, formally defined as

**Definition 4** (Extended Path, $epath$). *An* epath *is a path that has no repeated node along the path except that the starting node is allowed to be the ending node when the length of the path is larger than 2.*

For instance, $u_1 \to u_2 \to u_3 \to u_1$ is a valid *epath*, where $u_1$ is repeated at the ending node. Note that $u_1 \to u_2 \to u_1$ is not an *epath* since the length is not larger than 2.

A *component* of a graph is a connected subgraph that is not part of any larger connected subgraph [15]. Breadth-first search (BFS) [12] allows us to traverse all epaths of a component from a node. On the basis of epath and BFS [12], we define the term Truncated ePath Tree (TPT) as follows.

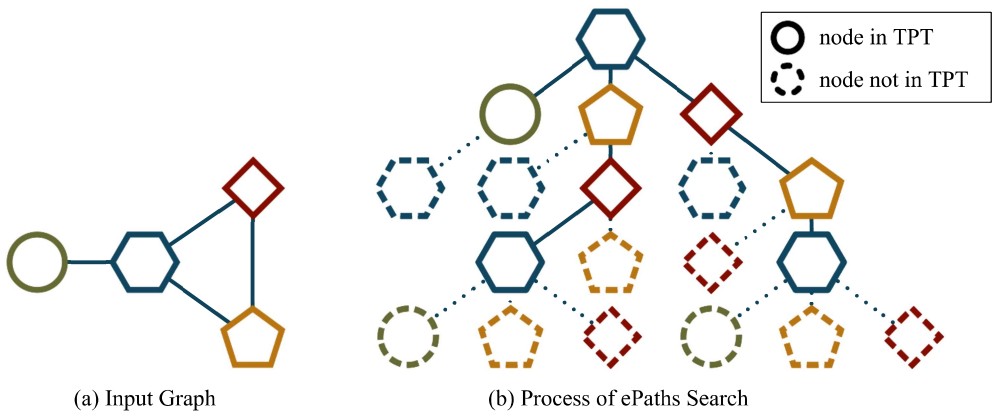

(a) Input Graph                                    (b) Process of ePaths Search

Figure 3: The height-4 Truncated ePath Tree of $G$ from $u$ (hexagon).

**Definition 5** (Truncated ePath Tree (TPT)). *Given a graph $G = (V, E)$ and a node $u \in V$, the height $k$ TPT $\mathrm{TPT}_{(G,u)}^k$ is an epath search tree obtained by running a BFS from $u$, where all epaths of length up to $k$ are accessed. Additionally, $u^{(l)}$ refers to a node $u$ at level $l$, and $\mathcal{C}^k(v)$ denotes the child node set of the node $v$ on level $k$.*

As shown in Figure 3, the original graph can form a height-4 truncated epath tree (TPT) as a tree structure of 8 nodes, where every tree path (from the root to the leaf) is an epath of length smaller than or equal to 4.

**Definition 6** (Truncated ePaths Forest (TPF)). *Given a graph $G = (V, E)$, the height $k$ TPF of $G$ is a multiset for all nodes $u \in V$ as*

$$\mathrm{TPF}_G^k \colon = \left\{ \left\{ \mathrm{TPT}_{(G,u)}^k \mid \forall u \in V \right\} \right\}.$$

**Redundancy-Free Graph Neural Network (RFGNN).** Assume there is a graph $G$, a $\mathrm{TPT} \in \mathrm{TPF}_G$ and $u \in \mathcal{V}(\mathrm{TPT})$ at level $(l-1)$. We denote $\mathbf{x}_u^{(l-1)}$ as a hidden feature transformed from the node features of $u$, $\mathbf{h}_v^{(l)}$ as a hidden feature of the subtree rooted at node $v \in \mathcal{C}^{(l-1)}(u)$, and $\mathbf{e}_{u,v}^{(l)}$ as a hidden feature transformed from the edge features of $(u, v)$. We can describe the RFGNN representation update process of a node $u$ at level $(l-1)$ as follows.

$$\mathbf{h}_u^{(l-1)} = \psi^{(l)} \left( \mathbf{x}_u^{(l-1)}, \cup_{v \in \mathcal{C}^{(l-1)}(u)} \phi^{(l)} \left( \mathbf{e}_{u,v}^{(l)}, \mathbf{h}_v^{(l)} \right) \right), \tag{7}$$

where $\cup$ denotes a differentiable and permutation invariant function, e.g., *sum* and *mean*, and $\phi$ and $\psi$ denote differentiable functions such as MLPs (Multi Layer Perceptrons).

To obtain the entire graph representation $\mathbf{h}_G$, we adopt a readout function $\chi$ that aggregates the root node representations of the TPTs of $G$:

$$\mathbf{h}_G = \chi \left( \left\{ \mathbf{h}_{root(\mathrm{TPT})}^{(0)} \mid \mathrm{TPT} \in \mathrm{TPF}_G \right\} \right), \tag{8}$$

where the readout function $\chi$ can be a simple permutation invariant function such as *sum*.

**Symmetries.** Given a graph $G$ with adjacency matrix $A$ and feature matrix $\mathbf{x}$, a function $f$ is (node) permutation equivalent if $f(A, \mathbf{x}) = f\left( MAM^T, M\mathbf{x} \right)$, for any permutation matrix $M$. GNN layers respect this equation, which ensures they compute the same functions up to a permutation (i.e., relabeling) of the nodes. Similarly, RFGNN is equivalent with respect to permutations of nodes, since the resulted TPTs and TPF is independent on any node permutation.

**Corollary 3.** *Redundancy-Free GNNs are permutation equivalent.*

**Memory consumption.** RFGNN's memory consumption may be higher than that of WL-test based GNNs since RFGNN requires constructing a Truncated ePath Forest (TPF). In future work, we will carefully study potential techniques, e.g., common paths consolidation, to overcome this issue.

## 4.2 The Expressive Power of RFGNN

1-WL is recognized as an indicator for the expressive power of GNNs [16, 17, 18]. In line with these work, we measure the expressive power of our model using a hashing procedure of TPFs. We first define the hashing procedure of TPFs (TPFH), Then, we prove that TPFH has the most expressive power if there is no height constraint on the height of TPFs and that the height $k$ TPFH is strictly more expressive than the 1-WL test of $k$ rounds.

**TPF Hashing (TPFH).** The height $k$ TPFH is a hashing scheme for height $k$ TPFs, which is a generalization of 1-WL [1]. In short, the hashing scheme is to assign labels to nodes. Let $c_u^{(l)}$ be the label assigned to $u^{(l)}$. And let $t_u^{(l)}$ be the label assigned to the tree rooted at $u^{(l)}$. We initialize the tree labels of nodes at level $k$ by their labels in the original graph. The hashing procedure is conducted recursively from the level $(k-1)$ to the root. Specifically, we collect the tree labels of the child nodes of $u$, denoted as $\{\{t_v^{(l+1)} \mid \forall v \in \mathcal{C}^{(l)}(u)\}\}$ (note it would be a null multiset if node $u$ has no child nodes). Then, we compute the tree label of $u$ by an injective mapping as $t_u^{(l)} = \mathrm{HASH}(c_u^{(l)}, \{\{t_v^{(l+1)} \mid \forall v \in \mathcal{C}^{(l)}(u)\}\})$, where HASH is a perfect hash function. Finally, we collect labels of all TPTs of a TPF as a multiset and injectively map it into a label by a perfect hash function. Two graphs are considered non-isomorphic if the labels of their TPFs are not equal. Otherwise, the test is inconclusive.

**Lemma 5.** *Non-isomorphic graphs will generate non-isomorphic TPFs if the height of TPFs is not constrained.*

If there were no height limitations on TPFs, the epath search would end when the TPF height reached the length of the longest epath. Then, the TPF of a graph would be composed of all of its epaths. Hashing TPFs with an unbounded height can therefore assign different labels to non-isomorphic graphs. In other words, TPFH without a height limitation offers the greatest expressive capability for distinguishing non-isomorphic graphs.

**Lemma 6.** *For two graphs $G_1$ and $G_2$, and two nodes $u \in \mathcal{V}(G_1)$ and $v \in \mathcal{V}(G_2)$, if the height $k$ WL subtree of $u$ is non-isomorphic to that of $v$, the height $k$ TPTs of them are non-isomorphic too.*

**Lemma 7.** *There exist two graphs $G_1$ and $G_2$ and two nodes, $u \in \mathcal{V}(G_1)$ and $v \in \mathcal{V}(G_2)$, where the height $k$ WL subtree of $u$ is isomorphic to that of $v$, but the height $k$ TPTs of them are non-isomorphic.*

According to Lemma 6 and Lemma 7, the expressive power of the height $k$ TPFH is better than that of the 1-WL with $k$ iterations. Similar to the proof in WL-test based GNNs [16], RFGNN can be shown to provide the same expressive power of TPFH when each layer of RFGNN are equipped with a sufficient number of layers and the parametric local aggregators they use can learn to be injective.

### 4.3 Advantages of RFGNN

We attempt to investigate what a RFGNN has learned by measuring how the output of a height $k$ RFGNN, as in Equation 7 , is influenced by input values and hidden representations.

**Lemma 8.** *Assume there is a height $k$ RFGNN as in Equation 7, a graph $G$, and an epath $ep$ in $G$ of length $l \leq k$. We can measure the **influence of the node feature** $\mathbf{x}_{ep[i]}^{(i)}$ **on the tree representation** $\mathbf{h}_{ep[0]}^{(0)}$, where $0 \leq i \leq l$, as*

$$\frac{\partial \mathbf{h}_{ep[0]}^{(0)}}{\partial \mathbf{x}_{ep[i]}} = \left( \prod_{j=0}^{i-1} \left( \partial_2 \psi^{(j)} \partial_2 \phi^{(j)} \right) \right) \left( \partial_{\mathbf{x}_{ep[i]}} \psi^{(i)} + \partial_2 \psi^{(i)} \partial_{\mathbf{x}_{ep[i]}} \phi^{(i)} \right).$$

*Similarly, we can measure the **influence of the tree representation** $\mathbf{h}_{ep[i]}^{(i)}$ **on the tree representation** $\mathbf{h}_{ep[0]}^{(0)}$, where $0 \leq i \leq l$, as*

$$\frac{\partial \mathbf{h}_{ep[0]}^{(0)}}{\partial \mathbf{h}_{ep[i]}^{(i)}} = \left( \prod_{j=0}^{i-1} \left( \partial_2 \psi^{(j)} \partial_2 \phi^{(j)} \right) \right) \partial_2 \psi^{(i)} \partial_{\mathbf{h}_{ep[i]}^{(i)}} \phi^{(i)}.$$

**Avoid over-quashing.** Lemma 8 points out that if all edges in the computational graph of a RFGNN is activated with the same probability, on average, epaths starting with the same node and having the same length would have the same influence on their shared starting node. This is similar to the finding of [19, 5]. Thus, the problem of over-squashing induced by message flow redundancy in the computational graph is avoided.

**Capture subgraphs at multiple levels of granularity.** An ideal method of graph similarity measurement should be able to capture subgraphs at different levels of granularity [11]. GIN [16] achieve this by an architecture similar to Jumping Knowledge Networks [20], which concatenates node representations across all layers. In contrast, RFGNN can directly detect the presence of subtrees of arbitrary heights of TPTs by deactivating information about trees (rooted nodes in TPTs) that do not belong to subtrees.

**Graph similarity comparison and graph overlap detection.** Intuitively, GNNs should be able to encode graphs with near graph edit distance (GED) [21] into similar representations. However, a similar graph pair's message flow sets created by WL-test based GNNs may vary significantly. This is because the difference in the message flow sets is dependent not only on the difference between the two graphs but also on their own topologies. In contrast, RFGNN will always yield message flow sets for graphs with near GED because of RFGNN's redundancy-free message passing, even if there are rings in the graphs, as stated by Lemma 9. Therefore, RFGNN is likely more capable than WL-test based GNNs at comparing graph similarity. A detailed discussion can be found in Appendix B.6.

**Lemma 9.** *Assume there is a graph $G = (V, E)$, a node $u \in V$, and an edge $e \in E$ within the $k$-hop neighborhood of $u$. Let $n$ be the number of rings in the $k$-hop neighborhood of $u$. We have that the number of repeations of $e$ within the height $k$ TPT of $u$ is no more than $1 + n$.*

## 5  Related Work

**Higher order GNNs.**  Recently, higher expressive GNNs have been developed, which based on enhanced WL algorithms. such as PPGN (based on $k$-FWL) [17], 1-2-3-GNN (based on $k$-WL) [22], CWNs (based on Cellular WL) [23], and Nested GNNs (based on Nested 1-WL) [24]. Nonetheless, none of them breaks the iterative aggregation-update scheme.

**Over-squashing issue.**  The *Jacobian* of a GNN-output is limited by a term given as the power of the graph normalised adjacency matrix [5]. And, [6, 5] both argued that small absolute values of the *Jacobian* are indicative of poor information propagation or over-squashing. Furthermore, [5] investigated graphs from a Ricci curvature perspective and demonstrated how negative curved edges contribute to the problem of over-squashing.

**Graph kernels.**  Graph kernels are frequently employed to measure the similarities between graphs by decomposing graphs into substructures and comparing these substructures [25]. The measurement of graph similarity could be utilized for subsequent tasks such as classifications [25]. In comparison to earlier graph kernels, Tree++ [7], which allows comparing graphs at multiple levels of granularity, has the highest classification accuracy on numerous benchmark datasets.

## 6  Experiments

We examine the performance of RFGNN and compare it with that of baseline approaches. RFGNN is implemented based on the Deep Graph Library (DGL) [26]. And the code is available at `https://github.com/RongqinChen/Redundancy-Free-Graph-Neural-Networks`.

First, we evaluate RFGNN by conducting a series of supervised graph classification tasks on 5 bioinformatics datasets (NCI1 [27], PTC [28], MUTAG [29], PROTEINS [30], ENZYMES [30]) and 3 small molecules datasets (BZR [31], COX2 [31], DHFR [31]). We adopted two different sorts of baselines: graph kernels (such as WL-subtree [3] and Tree++ [7]) and GNNs (such as GIN [16], PPGN [18], 1-2-3-GNN [22], and CWN [23]). We applied RFGNN with a height of 3. $\phi$ and $\psi$ functions in Equation 7 are an identify mapping (because because all these datasets have no edge attributes) and a 2-layer MLP, respectively. The experiment configurations follows that of GIN [16], such as the number of hidden units (32), batch size (128), 10-fold cross validation, the number of epochs (350), and learning rate (initialized as 0.01 decayed by 0.5 every 50 epochs). Configurations of baselines are follow their original papers. Due to the lack of executable codes, some Tree++ results are missing. In addition, a 1-2-3-GNN result is missing because of an out-of-memory error.

The data set statistics and the test set classification accuracy of RFGNN and baselines are shown in Table 1. We see that RFGNN outperforms GIN and even high order GNNs on most datasets, supporting our argument that RFGNN is a theoretically sound method for improving GNN performance.

Table 1: Statistics of the datasets and performance comparison.

| Datasets | MUTAG | PROTEINS | PTC | NCI1 | ENZYMES | BZR | COX2 | DHFR |
|---|---|---|---|---|---|---|---|---|
| #graphs | 188 | 1113 | 344 | 4110 | 600 | 405 | 467 | 756 |
| #classes | 2 | 2 | 2 | 2 | 6 | 2 | 2 | 2 |
| Avg. #nodes | 17.93 | 39.06 | 25.56 | 29.87 | 32.63 | 35.75 | 41.22 | 42.43 |
| Avg. #edges | 19.79 | 72.82 | 25.96 | 32.30 | 62.14 | 38.36 | 43.44 | 44.54 |
| TPF Avg. # nodes | 232.84 | 2165.3 | 336.56 | 376.48 | 1908.11 | 503.12 | 564.87 | 563.00 |
| TPF Avg. # edges | 216.77 | 2160.0 | 311.41 | 349.05 | 1904.98 | 469.98 | 525.87 | 522.69 |
| WL-subtree | $90.4 \pm 5.7$ | $75.0 \pm 3.1$ | $59.9 \pm 4.3$ | $\mathbf{86.0 \pm 1.8}$ | $62.6 \pm 1.2$ | $87.3 \pm 0.8$ | $81.2 \pm 1.1$ | $82.4 \pm 0.9$ |
| Tree++ | – | $75.5 \pm 0.5$ | – | $85.8 \pm 0.1$ | – | $87.9 \pm 1.0$ | $84.3 \pm 0.9$ | $83.7 \pm 0.6$ |
| GIN [16] | $90.0 \pm 8.8$ | $76.2 \pm 2.8$ | $64.6 \pm 7.0$ | $82.7 \pm 1.7$ | $72.5 \pm 6.1$ | $89.9 \pm 3.8$ | $\mathbf{87.6 \pm 4.0}$ | $82.9 \pm 3.9$ |
| 1-2-3-GNN [22] | $92.2 \pm 4.4$ | $77.3 \pm 4.2$ | $67.6 \pm 5.5$ | $61.0 \pm 5.3$ | – | $82.0 \pm 3.3$ | $78.7 \pm 1.3$ | $63.4 \pm 4.9$ |
| PPGN [17] | $90.6 \pm 8.7$ | $77.2 \pm 4.7$ | $66.2 \pm 6.5$ | $83.2 \pm 1.1$ | $63.3 \pm 11.8$ | $89.4 \pm 4.0$ | $86.7 \pm 4.1$ | $79.6 \pm 4.5$ |
| CWN [23] | $92.7 \pm 6.1$ | $77.0 \pm 4.3$ | $68.2 \pm 5.6$ | $83.6 \pm 1.4$ | $62.6 \pm 8.1$ | $78.8 \pm 1.0$ | $78.4 \pm 1.1$ | $68.1 \pm 3.1$ |
| RFGNN | $\mathbf{93.3 \pm 6.0}$ | $\mathbf{77.5 \pm 3.4}$ | $\mathbf{71.5 \pm 3.2}$ | $83.6 \pm 1.6$ | $\mathbf{75.3 \pm 5.0}$ | $\mathbf{90.9 \pm 3.2}$ | $87.4 \pm 4.6$ | $\mathbf{84.5 \pm 4.6}$ |

To further evaluate the ability to model long-range interactions and the capacity to quantify graph similarity of RFGNN, we computed and compared the performance of RFGNNs with heights ranging from 6 to 8 and baseline approaches on QM9 [32, 33], a graph dataset with 12 regression tasks. QM9 [32, 33] contains 130K small molecules. The task here is to separately perform regressions on 12 targets representing energetic, electronic, geometric, and thermodynamic properties, based on the graph structure and node/edge features. All evaluation metrics are mean absolute error (MAE). We use a *sum* aggregator to obtain graph representations. $\phi^{(l)}$ and $\psi^{(l)}$ in Equation 7 both are 2-layer MLPs. All other settings follow [24].

We report the average test MAE across 10 folds and show the QM9 results in Table 2. The results of baseline methods are extracted from [24]. We highlight the results of RFGNNs if they are better than those of baselines. Several positive conclusions can be drawn from the experimental results: (1) RFGNNs achieved the best results on 7 out of 12 prediction targets, which means RFGNNs have better performance on measuring graph similarity. (2) The performance of RFGNN increases with increasing height on several targets, which underlines the need for modeling long-range interactions in these targets. (3) Ne.1-GNN has the potential to model longer-range interactions better than RFGNN-8 due to its deeper layers; however, Ne.1-GNN does not perform better than RFGNN-8. We believe that redundancy-free message passing allows RFGNN to model long-range interactions more competently than Ne.1-GNN, as Ne.1-GNN's message passing is redundant.

Table 2: MAE results on QM9 (smaller the better)

| Target | $\mu$ | $\alpha$ | $\varepsilon_{\text{HOMO}}$ | $\varepsilon_{\text{LUMO}}$ | $\Delta\varepsilon$ | $\langle R^2 \rangle$ | ZPVE | $U_0$ | $U$ | $H$ | $G$ | $C_v$ |
|---|---|---|---|---|---|---|---|---|---|---|---|---|
| DTNN [34] | **0.244** | 0.95 | 0.00388 | 0.00512 | 0.0112 | 17.0 | 0.00172 | 2.43 | 2.43 | 2.43 | 2.43 | 0.27 |
| MPNN [2] | 0.358 | 0.89 | 0.00541 | 0.00623 | 0.0066 | 28.5 | 0.00216 | 2.05 | 2.00 | 2.02 | 2.02 | 0.42 |
| Deep LRP [35] | 0.364 | 0.298 | 0.00254 | 0.00277 | 0.00353 | 19.3 | 0.00055 | 0.413 | 0.413 | 0.413 | 0.413 | 0.129 |
| 1-GNN [22] | 0.493 | 0.78 | 0.00321 | 0.00355 | 0.0049 | 34.1 | 0.00124 | 2.32 | 2.08 | 2.23 | 1.94 | 0.27 |
| 1-2-GNN [22] | 0.493 | 0.27 | 0.00331 | 0.00350 | 0.0047 | 21.5 | 0.00018 | **0.0357** | 0.107 | 0.070 | 0.140 | 0.0989 |
| 1-3-GNN [22] | 0.473 | 0.46 | 0.00328 | 0.00354 | 0.0046 | 25.8 | 0.00064 | 0.6855 | 0.686 | 0.794 | 0.587 | 0.158 |
| 1-2-3-GNN [22] | 0.476 | 0.27 | 0.00337 | 0.00351 | 0.0048 | 22.9 | 0.00019 | 0.0427 | 0.111 | **0.0419** | **0.0469** | 0.0944 |
| Ne.1-GNN [24] | 0.428 | 0.29 | 0.00265 | 0.00297 | 0.0038 | 20.5 | 0.00020 | 0.295 | 0.361 | 0.305 | 0.489 | 0.174 |
| Ne.1-2-GNN [24] | 0.437 | 0.278 | 0.00275 | 0.00271 | 0.0039 | 20.4 | 0.00017 | 0.252 | 0.265 | 0.241 | 0.272 | 0.0891 |
| Ne.1-3-GNN [24] | 0.436 | 0.261 | 0.00265 | 0.00269 | 0.0039 | 20.2 | 0.00017 | 0.291 | 0.278 | 0.267 | 0.287 | 0.0879 |
| Ne.1-2-3-GNN [24] | 0.433 | 0.265 | 0.00279 | 0.00276 | 0.0039 | 20.1 | **0.00015** | 0.205 | 0.200 | 0.249 | 0.253 | 0.0811 |
| RFGNN-6 | 0.323 | **0.216** | **0.00204** | **0.00206** | **0.00311** | **14.277** | 0.000156 | 0.0997 | 0.1257 | 0.1153 | 0.1103 | **0.07720** |
| RFGNN-7 | 0.318 | **0.209** | **0.00200** | **0.00205** | **0.00310** | **14.316** | 0.000162 | 0.1082 | 0.1149 | 0.1101 | 0.1099 | **0.07420** |
| RFGNN-8 | 0.322 | **0.208** | **0.00200** | **0.00205** | **0.00314** | **14.057** | 0.000159 | 0.0775 | **0.0741** | 0.0744 | 0.1015 | **0.07485** |

## 7 Conclusions

In this work, we carefully investigated the message flow redundancy issue in recent GNNs and its implications. In particular, we have proved that message flow redundancy is one key factor in the over-squashing phenomenon. To address the message flow redundancy issue, we have proposed Redundancy-Free Graph Neural Networks (RFGNN), which conducts redundancy-free message passing. Theoretically, we demonstrate the benefits of our method, and experimentally, we show that our RFGNN outperforms existing WL-test based GNNs on a large number of graph property prediction challenges.

## 8 Acknowledgement

The authors appreciate the constructive comments and valuable suggestions from the reviewers. This work was supported by the National Key Research and Development Plan of China (No.2019YFB2102100), the Strategic Priority Research Program of Chinese Academy of Sciences (Grant No. XDB38040200), the National Key Research and Development Plan of China (No.2021YFF0901104), the Science and Technology Development Fund Macau (SKL-IOTSC-2021-2023, 0015/2019/AKP, 0031/2022/A), University of Macau (MYRG2019-00119-FST, MYRG2022-00252-FST), Wuyi University Hong Kong and Macau joint Research Fund (2021WGALH14).

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
