# OpenReview forum: "Redundancy-Free Message Passing for Graph Neural Networks"
_NeurIPS.cc/2022/Conference — NeurIPS 2022 Accept_

### Official Review · Reviewer_hcxm · 2022-07-07

**Rating:** 7
**Confidence:** 4
**Soundness:** 4 excellent
**Presentation:** 3 good
**Contribution:** 3 good

**Summary:**

The authors aim to theoretically analyze the redundancy in GNNs and its practical implications, which is an interesting and important problem in the GNN community. To overcome the redundancy problem, the authors propose an expressive RFGNN based on the so-called Truncated ePath Tree. Experiments demonstrate that RFGNN outperforms existing models.

**Questions:**

1. Whether RFGNN is applicable to larger graphs such as ogbg-ppa and ogbg-code2?
2. Since one of the motivations to alleviate over-squashing is to better model long-distance interactions, I suggest the authors to conduct experiments to evaluate the effect of the number of GNN layers. The current number of GNN layers is always 3, which may not reflect“long-distance interactions”.
3. In Line 182, the authors claim that “the influence along a long path is always less than that of its sub-paths”. Is there any possibility that distant nodes should contribute less than closer nodes?

**Limitations:**

The authors have addressed the limitations of their work, but did not discuss the potential negative societal impact.

**Strengths And Weaknesses:**

- Originality
  - The authors are the first to theoretically analyze the redundancy GNNs and its practical implications. Thus, the submission is certainly an original work.
  - The idea that using extended path to develop a redundancy-free message passing paradigm seems novel to me.
- Quality
  - This paper is well motivated. The authors did a good job in illustrating the redundancy problem in message passing and propose a reasonable solution to it.
  - This paper is technically solid. The authors rigorously analyze why redundancy causes message over-squashing. Besides, the authors provide theoretical analysis for the proposed method and show their method outperforms 1-WL test.
  - My major concern is the scalability of the proposed RFGNN. As shown in Table 1, in TPF, the average number of nodes/edges becomes at least ten times larger than that in the original graph. I am wondering if RFGNN is applicable to larger graphs such as ogbg-ppa and ogbg-code2.
- Clarity
  - The writing of this paper is overall clear. My only complaint is about Figure 1. It looks confusing to me and the authors may want to give a more clear illustration in either the main text or caption.
- Significance
  - Both the theoretical and empirical results are significant.

---

> ### Author Response · Authors · 2022-08-02
> **Responses to Reviewer hcxm**
>
> We are happy to note the reviewer has appreciated the solidity of the theoretical contribution. We respond to all the outstanding comments and questions below.
>
> *W1: "My major concern is the scalability of the proposed RFGNN. As shown in Table 1, in TPF, the average number of nodes/edges becomes at least ten times larger than that in the original graph. I am wondering if RFGNN is applicable to larger graphs such as ogbg-ppa and ogbg-code2"\
> Q1: "Whether RFGNN is applicable to larger graphs such as ogbg-ppa and ogbg-code2?"*
>
> **Response:** We agree with the reviewer that the space complexity of RFGNN is a concern in graph learning tasks, which is O(n^2) (where n indicates the number of nodes) as mentioned in Appendix D. Indeed, RFGNN requires more resources (time and memory) to run on larger graphs, such as ogbg-ppa and ogbg-code2. Given the limited time and resources in hand, we are not able to report the result of these two datasets in the rebuttal phase.
>
> However, we would like to emphasize a key point that the memory requirement of RFGNN is smaller than that of other graph neural networks dedicated to achieving high expressive power, such as PPGN [1] and higher-order GNN [2]. Besides, the time complexity of RFGNN is also smaller than that of these methods.
>
> Specifically, assuming a graph with n nodes, each layer of the high-order GNN has O(n^3) space complexity (number of subsets) and O(n^4) time complexity due to O(n^3) subsets and O(n) neighbors. Each layer of PPGN has O(n^3) space complexity as it stores only the second order tensors and O(n^3) time complexity due to the matrix multiplication. However, our RFGNN requires only O(n^2) space complexity due to the size of TPF (each TPT has at most 2n nodes, double is for there may be repeated nodes in ePath) and O(n^3) time complexity (since computation happens on edges which is O(n^2)). The detail complexity analysis has been given in Appendix D.
>
> [1]: Maron, H., Ben-Hamu, H., Serviansky, H., & Lipman, Y. (2019). Provably Powerful Graph Networks. *ArXiv, abs/1905.11136*.\
> [2]: Li, P., & Leskovec, J. (2022). The Expressive Power of Graph Neural Networks. *Graph Neural Networks: Foundations, Frontiers, and Applications*.
>
> *W2: "The writing of this paper is overall clear. My only complaint is about Figure 1. It looks confusing to me and the authors may want to give a more clear illustration in either the main text or caption."*
>
> **Response:** We apologize that Figure 1 is not clear enough. We have revised Figure 1 and updated the explanatory text to improve its readability. Please check the new figure in our revised submission.
>
> *Q2: "Since one of the motivations to alleviate over-squashing is to better model long-distance interactions, I suggest the authors to conduct experiments to evaluate the effect of the number of GNN layers. The current number of GNN layers is always 3, which may not reflect 'long-distance interactions'".*
>
> **Response:** We thank the reviewer for suggesting more experiments. To evaluate the effect of the number of GNN layers (and height of RGNN), we compared RFGNN with GIN at different layers, the performance is shown in the Appendix (Table 3). As shown in the table, the performance of RFGNN on most datasets is stable as the height increased. This shows that RFGNN of height 3 is sufficient to capture the structure of the evaluated graphs.
>
> To further evaluate the ability of RFGNN to model long-range interactions, we added a new set of experiments. In these experiments, we apply RFGNNs of heights from 6 to 8 on the QM9 dataset with 12 regression tasks. The results are shown in the Appendix (Table 4).
>
>
> *Q3: "In Line 182, the authors claim that 'the influence along a long path is always less than that of its sub-paths'. Is there any possibility that distant nodes should contribute less than closer nodes?"*
>
> **Response:** We thank the reviewer for this question. Yes, we claim that the influence along a long path is always less than that of its sub-paths. In our work, Corollary 2 indicates the relative influence along a trail is proportional (in terms of probability) to the number of message flows derived from the trail. Moreover, Lemma 2 indicates that a trail of longer length usually yields fewer messaging paths than a child trail of shorter length. Consequently, the influence along a long path is always less than that of its sub-paths. From another perspective, the message flows transmitting the information of a long path transmit the information of its sub-paths too, which serves as another explanation for this conclusion.

---

### Official Review · Reviewer_7sMW · 2022-07-11

**Rating:** 7
**Confidence:** 3
**Soundness:** 4 excellent
**Presentation:** 3 good
**Contribution:** 4 excellent

**Summary:**

One challenge of conventional GNNs is the propagation of information between long-distance node / long-length paths. This paper argues the redundant aggregation of the same node/edge information attenuates the effect of features coming from long distance paths (Message Over-squashing).
As a solution, Redundancy-Free Graph Neural Networks (RFGNN) are proposed. They are a variation of GNNs which is  redundancy-free.
This contribution also proposes a theoretical framework allowing to compare RFGNN with traditional WL-GNNs. In particular, it is shown that:
- The expressive powers of RFGNN can surpass the conventional GNNs (TPF Hashing is more powerful than WL-1).
- Over-squashing is reduced by de-redundancy.
Eventually, RFGNN is evaluated on graph classification tasks against several baselines (including GIN, k-GNN, and CIN) and on several bioinformatics datasets (NCI1, PTC, MUTAG, PROTEINS, ENZYMES) as well as 3 small molecules datasets (BZR, COX2, DHFR).

**Questions:**

- The TPF Hashing (TPFH) is compared to WL-1 test. Why not compare a TPFH with trees of height k with a WL-k test? Would they be the same?

- Shouldn’t k-Tree RFGNN be as powerful as a k-order GNN when learning is done appropriately?

- What is the blue color in the result Table? It is a bit misleading because RFGNN is the 3rd score and not the 2nd one.


**Limitations:**

Limitations are not discussed. No potential negative societal impact.

**Strengths And Weaknesses:**

## Strengths
 - I believe this work is very novel. It introduces a new type of Graph Neural Network as well as theoretical framework to analyze them.
 - The contribution is well written and of high quality. Given the space constraint, the contribution is relatively clear.
 - It seems that RFGNN achieves superior performance in graph discrimination than traditional GNNs and even k-order GNNs.

 ## Weaknesses
  - I believe that NeurIPS is not a good format for this contribution. 9 pages are just not enough to fit all the relevant information and the presentation of this paper suffers a bit from this. Therefore, if the paper is accepted, I would suggest to the authors to publish a longer version on arxiv that includes some of the appendices in the main text as well as spend a bit more time explaining the core idea of this contribution. This being said, I agree that such a contribution should be presented at NeurIPS.
  - The experimental evaluation feels a bit weak. It misses some scores (a lot of "-"). A longer discussion of the results would be welcome. According to me, not all baselines are equivalent and the main one should be the k-order GNN.
  - According to me, implementing the RFGNN is not trivial. Some explanation on how it is done would be welcome. Code is provided, though.

---

> ### Author Response · Authors · 2022-08-02
> **Responses to Reviewer 7sMW**
>
> We thank the reviewer for their thoughtful, comprehensive, and constructive review as well as for acknowledging our theoretical analyses and empirical evaluations.
>
> *W1: I believe that NeurIPS is not a good format for this contribution. 9 pages are just not enough to fit all the relevant information and the presentation of this paper suffers a bit from this. Therefore, if the paper is accepted, I would suggest to the authors to publish a longer version on arxiv that includes some of the appendices in the main text as well as spend a bit more time explaining the core idea of this contribution. This being said, I agree that such a contribution should be presented at NeurIPS.*
>
> **Response:** Thanks for recognizing the value of our work and giving encouraging comments. If our paper is fortunate enough to be accepted in NeurIPS, we will upload an extended version to arxiv (including the restructure of the paper) that will better convey our theoretical findings to public. In addition, to better demonstrate the empirical evaluation of our method, we will report the performance of RFGNN on more datasets in the extended version.
>
> *W2. "The experimental evaluation feels a bit weak. It misses some scores (a lot of "-"). A longer discussion of the results would be welcome. According to me, not all baselines are equivalent and the main one should be the k-order GNN."\
> Q2. "Shouldn’t k-Tree RFGNN be as powerful as a k-order GNN when learning is done appropriately?"*
>
> **Response:** Thank you for the suggestion. Yes, we agree with you that the experimental evaluation should be enhanced, and this concern is also raised by other reviewers. Given the limit time of the rebuttal phase, we have tried our best to fill the missing results in Table 1. Some experiments are still missing since the format of some datasets cannot be fit into the structure of the baseline models. We need more time to prepare the experiment evaluation for these methods.
>
> Regarding the k-order GNNs, we have compared with 1-2-3 GNN and PPGN. Please be noted that the advantages of our RFGNN are not only higher expressive power but also the ability to measure the similarity between graphs in different granularity. Thereby, we do not only focus on those "high expressive power" models but also consider those "structure-aware" models, e.g., Tree++.
>
> *W3. "According to me, implementing the RFGNN is not trivial. Some explanation on how it is done would be welcome. Code is provided, though."*
>
> **Response:** Thank you for your constructive comments. We will add the implementation detail in our Github repository after this work has been accepted.
>
> *Q1. "The TPF Hashing (TPFH) is compared to WL-1 test. Why not compare a TPFH with trees of height k with a WL-k test? Would they be the same?"*
>
> **Response:** Thank you again for the insightful comment. To the best of our understanding, the height in TPFH and the order in WL test are two different concepts in the graph learning process. In RFGNN, we can increase the height of TPF to catch long-distance neighbors. In WL-k test, the increment of the WL-test order is to capture higher order substructures (like 3-tuple nodes, 4-tuple nodes, and rings). Moreover, the space and time complexity of these two approaches are also different. Each iteration of WL-k requires O(n^k) space (=the number of subsets) and O(n^{k+1}) time (=O(n^k) subsets with O(n) neighbors). RFGNN only uses O(n^2) space and O(n^3) time to compute an iteration. In our experiments, we can run experiments of RFGNN up to height 8 but it is infeasible to run WL-8 test on moderate-size graphs. According to the experiments in [1] and [2], the WL-test based methods usually set k to a small value (e.g., 3) in practices.
>
> [1]: Morris, C., Ritzert, M., Fey, M., Hamilton, W.L., Lenssen, J.E., Rattan, G., & Grohe, M. (2019). Weisfeiler and Leman Go Neural: Higher-order Graph Neural Networks. ArXiv, abs/1810.02244.\
> [2]: Maron, H., Ben-Hamu, H., Serviansky, H., & Lipman, Y. (2019). Provably Powerful Graph Networks. ArXiv, abs/1905.11136.
>
> *Q3: "What is the blue color in the result Table? It is a bit misleading because RFGNN is the 3rd score and not the 2nd one."*
>
> **Response:** Thank you for pointing out this typo. We have fixed this in the revised version.

---

> > ### Comment · Reviewer_7sMW · 2022-08-08
> > **Answer to the response**
> >
> > Thank you for your response. I am satisfied with your answer.
> >
> > I have also read the other reviews. I could not find anything that would make me change drastically my opinion about the paper. Most of us agree that the experimental section could be more exhaustive. Still, I believe that this contribution is mostly theoretical. I have not read the proofs (but theorems seems sound). Unless, someone finds an important mistake in the proofs, I suggest the acceptance of this paper.

---

### Official Review · Reviewer_Da5u · 2022-07-11

**Rating:** 6
**Confidence:** 3
**Soundness:** 3 good
**Presentation:** 3 good
**Contribution:** 3 good

**Summary:**

This work aims to eliminate the message flow redundancy in traditional message passing schemes so as to reduce the over-squashing issue. The authors first explain the redundancy in the computational graph of message passing and then theoretically quantify that the redundancy would cause the over-squashing issue. Based on these findings, the authors propose a message-passing GNN that is message flow redundancy-free and theoretically has better expressive power than 1-WL.

To empirically evaluate the performance of the proposed GNN (RFGNN), semisupervised graph classification tasks are done to compare RFGNN to several baselines. The results show that RFGNN in general exhibits better performance than the baselines.

**Questions:**

I saw the height of the Truncated ePaths Forest (TPF) in RFGNN is 3 in experiments, how would the height affect the performance? Besides,  I'm not sure why the length of each epath should be larger than 2. Will the choice of 2 affect any analysis and the performance?

**Limitations:**

Since the proposed GNN is redundancy-free for the message flows, it would be interesting to investigate the computational complexity of the proposed model. If the model can reduce redundant computations and save memory/time, that can be also an advantage to be discussed.

**Strengths And Weaknesses:**

Strengths:
1. The investigation of the message flow redundancy in message passing together with the over-squashing problem is novel. There are existing works related to the redundancy in message passing. However, they usually focus on reducing the computational complexity caused by redundant operations.
2. The authors conduct theoretical analysis when quantifying the effect of over-squashing and analyzing the expressive power of the proposed GNN. These analyses make the proposed method more solid.
3. The paper is generally easy to follow and well-written. The concepts and the proposed models are clearly explained.

Weaknesses:
1. The experiments are relatively weak compared to the theoretical part. The experiments need to be better designed to better demonstrate the performance of the proposed method. For example, the performance difference between the proposed RFGNN and the other baselines can be smaller than the stds as shown in Table 1, it would be necessary to confirm the statistical significance when comparing them. Besides, the results of GIN. k-order GNN, and CIN are not computed for Enzymes, BZR, Cox2 and DHFR in Table 1, thus the comparison is not comprehensive enough.
2. There are existing works investigating the redundant computations in GNNs (e.g. Redundancy-Free Computation for Graph Neural Networks, KDD 2020). The authors should address these works in the Related Works section.

---

> ### Author Response · Authors · 2022-08-02
> **Responses to Reviewer Da5u**
>
> We thank the reviewer for their insightful review. Please be noted that our research aims to provide a redundancy-free message passing paradigm for GNNs, which is orthogonal to those work aiming at providing redundancy-free computations for GNNs. For the other comments, we have tried our best to address them in the response. In the revised submission, we have enhanced our experiments by running baselines on more datasets (Table 1) and adding the discussion to the redundancy free computation for GNNs into the related work. Please find the detailed responses as follows.
>
> *W1. "The experiments are relatively weak compared to the theoretical part. The experiments need to be better designed to better demonstrate the performance of the proposed method. For example, the performance difference between the proposed RFGNN and the other baselines can be smaller than the stds as shown in Table 1, it would be necessary to confirm the statistical significance when comparing them. Besides, the results of GIN. k-order GNN, and CIN are not computed for Enzymes, BZR, Cox2 and DHFR in Table 1, thus the comparison is not comprehensive enough."*
>
> **Response:** Thank you for pointing this issue out. Given the limit time of the rebuttal phase, we have tried our best to run the experiments on those missing results in Table 1. Some experiments are still missing since the format of some datasets cannot be fit into the structure of the baseline models. We need more time to prepare the experiments for these datasets.
>
> *W2. "There are existing works investigating the redundant computations in GNNs (e.g. Redundancy-Free Computation for Graph Neural Networks, KDD 2020). The authors should address these works in the Related Works section."*
>
> **Response:** First, we agree that there are existing work investigating the redundancy issue in GNNs. However, we believe our redundancy issue is different from the redundancy issue in “Redundancy-Free Computation for Graph Neural Networks”. In the KDD20 paper, their objective is to remove the repeated and inefficient “redundant computations” in a GNN computational graph. Their solution is based on an observation that neighbors are commonly shared by more than one nodes (as shown in Figure 1 in that paper). Conventional GNNs perform equivalent processes on those nodes who share common neighbors.
> Orthogonally, our study focuses on removing redundant message flows in a GNN computational graph rather than the redundant computation operations. Note that removing redundant computations does not secure a redundancy-free message flows since the same node may still appear in the computational graph more than one times. To the best of our understanding, we are the first to theoretically investigate the redundancy issue in the message passing of GNNs and their practical ramifications.
>
> *Q1: "I saw the height of the Truncated ePaths Forest (TPF) in RFGNN is 3 in experiments, how would the height affect the performance? Besides, I'm not sure why the length of each epath should be larger than 2. Will the choice of 2 affect any analysis and the performance?"*
>
> **Response:** We have included the result of TPF by varying the height of the trees in the Appendix section. The result is not sensitive to the height of TPF in our evaluated datasets since the expressive power of TPF of height 3 is already sufficient to capture the subgraph patterns in our evaluated datasets, which is similar to the typical number of layers in other GNN solutions.
>
> Regarding the length constraint of epath, it is only applicable to those epaths if their starting node is identical to the ending node. In other words, we do not allow self-loop epath of length 1. The length of an epath could be 0 (the root node is isolated), 1 (the epath from hexagon node to circular node in Figure 2), 2 (if the circular node in Figure 2 has another neighbour node, denoted x, the path from hexagon node to x passing circular node is a valid epath), or even longer.
>
> Please be also noted that our intent of this length constraint in epath is to represent a ring structure, as described in line 60.
>
> *L1: "Since the proposed GNN is redundancy-free for the message flows, it would be interesting to investigate the computational complexity of the proposed model. If the model can reduce redundant computations and save memory/time, that can be also an advantage to be discussed."*
>
> **Response:** We have already provided the complexity analysis in Section D of Appendix. We briefly summarize our result as follows. The space complexity and time complexity of RFGNN are O(n^2) and O(n^3), respectively, which are lower than those of k-order GNNs [1].
> [1]: Li, P., & Leskovec, J. (2022). The Expressive Power of Graph Neural Networks. Graph Neural Networks: Foundations, Frontiers, and Applications.

---

> > ### Comment · Reviewer_7sMW · 2022-08-08
> > **Comment about weakness #2.**
> >
> > During my review, I have checked the paper (Redundancy-Free Computation for Graph Neural Networks, KDD 2020). While the title is (unfortunately) very similar, I agree with the authors that it is a very different line of work. Therefore, I do not see the missing discussion of that reference as a weakness. I am unsure if the authors should mention that their work is different in order to avoid the confusion of the reader.

---

> > ### Comment · Reviewer_Da5u · 2022-08-09
> > **Comment on authors' responses**
> >
> > After reading the authors' responses, I agree that the KDD paper focuses on another issue. If the page limit is enough, I would suggest briefly discriminating the work. Besides, given the updated results in Table 1 and the theoretical contribution of the work, I'm satisfied and have raised my rating.

---

### Official Review · Reviewer_6V6A · 2022-07-12

**Rating:** 5
**Confidence:** 4
**Soundness:** 2 fair
**Presentation:** 3 good
**Contribution:** 3 good

**Summary:**

This paper analyzes the redundancy problem in message passing in graph neural networks, and proposes a method to address this issue. Theoretical analysis and empirical verification are presented to justify the effectiveness of the proposed method.
The contributions can be summarized as follows:
1) Formulate and analyze the redundancy problem in message flows and suggest its impact on hindering the performance of GNNs.
2) Propose a redundancy-free GNN (RFGNN), and justify the superior expressive power of RFGNN in terms of addressing the over-squashing issue.
3) Show another advantage of RFGNN, i.e., capable of measuring the similarity between graphs in different granularity and detecting overlapping subgraphs.
4) Empirically verify the effectiveness of RFGNN through graph classification tasks on eight benchmark datasets by comparing with graph kernels and powerful GNNs.

**Questions:**

Questions:
1. As the paper described in section 6 (page 9: line 314),  the experiment is semi-supervised graph classification. But the experiment follows the configuration of <How powerful are Graph Neural Networks>, which conducts experiments with supervised graph classification tasks. Can you explain this inconsistency? And give more details of the semi-supervised learning in this experiment?

2. Why are most baseline results missing in Table 1?

**Limitations:**

1. No appendix with proofs.

2. It is preferable to give instances during the theoretical analysis, so the readers can follow the idea easier.

3. The results in Table 1 are incomplete.

4. More experiments are needed to demonstrate the details of the learning of RFGNN.

**Strengths And Weaknesses:**

Strengths:
1. This paper studies an important problem, i.e., the redundancy within the message passing flow, which is highly related to the performance of graph-level tasks where higher-order neighbors are important.

2. This paper is well organized, and the theoretical analysis is clearly presented. The redundancy problem is introduced and formulated strictly in the research context.

3. This paper also analyzes the negative impact of message flow redundancies and connects it with the over-squashing phenomenon. The significance of this work is highlighted by showing its ability to address the over-squashing problem.

4. The expressive power of RFGNN is studied by 1) theoretical analysis and comparison with 1-WL, 2) empirical verification through experiments by comparing with graph kernels and power GNNs such as GIN, k-GNN, and CIN.

5. The proposed method shows other advantages in measuring graph similarity and detecting overlapping subgraphs.


Weakness:
1. It is said at the beginning of section 3 that “all proofs can be found in Appendix“, but there is no appendix attached.

2. The lack of illustrations/visualization during the analysis in Section 3 makes it difficult to catch the idea. For instance, on page 4, lines 133-139, this part can be illustrated with an example.

3. The results for baselines in Table 1 are reported from the original papers. However, the different configurations may result in different performance results of the baselines. In this paper, the experiment follows the configuration in the paper <How powerful are Graph Neural Networks>. The configurations between RFGNN and other baselines (e.g., Tree++) are not consistent.

4. The experiments are kind of simple. Only one graph classification experiment is presented. There are only five baseline methods, and around half of the baseline results are not available. And there are no explanations for the missing results.

---

> ### Author Response · Authors · 2022-08-02
> **Responses to Reviewer 6V6A**
>
> We thank the reviewer for the insightful review. First, we want to clarify that the Appendix section (including the proofs and supplementary experiments) was uploaded as a supplementary material file on 26 May. Second, we have tried our best to address the concerns, including filling the missing results in Table 1 and supplementing the experiment with a graph regression with 12 targets. The detailed responses are as follows.
>
> *W1: "It is said at the beginning of section 3 that ‘all proofs can be found in Appendix’, but there is no appendix attached."\
> L1. "No appendix with proofs."*
>
> **Response:** The Appendix section was uploaded as a supplementary material (rfgnn_appendix.pdf) on 26 May, that includes the proofs and supplementary experiments.
>
> *W2: "The lack of illustrations/visualization during the analysis in Section 3 makes it difficult to catch the idea. For instance, on page 4, lines 133-139, this part can be illustrated with an example."\
> L2. "It is preferable to give instances during the theoretical analysis, so the readers can follow the idea easier."*
>
> **Response:** In the revised version, we have added an example to explain the analysis of the number of message flows and revised the description to make it easier for the reader to understand. Moreover, we have revised Lemma 2 that fixes the over-estimation in our last version.
>
> *W3: "The results for baselines in Table 1 are reported from the original papers. However, the different configurations may result in different performance results of the baselines. In this paper, the experiment follows the configuration in the paper. The configurations between RFGNN and other baselines (e.g., Tree++) are not consistent."*
>
> **Response:** We follow the experiment configurations of GIN, such as the number of hidden units (32), the batch size (128), 10-fold cross validation, and learning rate (initialized as 0.02 decayed by 0.5 every 50 epochs). The only difference in configuration is the height of RFGNN and the number of layers of GIN. Because the expressive capacity of TPF at a height of 3 is already sufficient to capture the subgraph patterns, we employed RFGNN with a height of 3. The results for baselines in Table 1 are reported from the original papers since we believe these are the best reported results of the baselines.
>
> *Q1: "As the paper described in section 6 (page 9: line 314), the experiment is semi-supervised graph classification. But the experiment follows the configuration of, which conducts experiments with supervised graph classification tasks. Can you explain this inconsistency? And give more details of the semi-supervised learning in this experiment?"
> Response: It is a typo in the writing. We only conducted supervised graph classification tasks in this experiment.\
> L3. "The results in Table 1 are incomplete."\
> Q2: "Why are most baseline results missing in Table 1?"*
>
> **Response:** At the NeurIPS submission deadline, we fully focused on our own method, including the experiment evaluation and theorical analysis but not the other baseline methods. For the missing results, it is caused by the format of some datasets that cannot be fitted into the structure of the baseline models. However, we agree with you and other reviewers that the experimental evaluation should be enhanced. Given the limit time of the rebuttal phase, we have tried our best to fill in the missing results in Table 1.
>
> *W4. "The experiments are kind of simple. Only one graph classification experiment is presented. There are only five baseline methods, and around half of the baseline results are not available. And there are no explanations for the missing results."\
> L4. More experiments are needed to demonstrate the details of the learning of RFGNN.*
>
> **Response:** We have reported a graph regression experiment on QM9 which has 12 prediction tasks. Due to the space limitation, we can only put the results in the Appendix (in our last submitted version). Nevertheless, we agree that we should include more experiments to demonstrate the details of the learning of RFGNN. The newly added experiments are described below.
> - To evaluate the effect of the number of GNN layers (and height of RGNN), we have added a set of experiments, where we compared RFGNN with GIN at different layers. The performance is displayed in the Appendix.
> - To further evaluate the ability of RFGNN to model long-range interactions, we have added a new set of experiments. In these experiments, we apply RFGNNs of heights from 6 to 8 on the QM9 dataset with 12 regression tasks.

---

### Meta-Review · Area_Chair_WmcR · 2022-08-24

**Recommendation:** Accept
**Confidence:** Certain

**Metareview:**

This work focuses on a well-known (inductive) bias of message-passing: the fact that information from far away neighbors is dilluted and taken less into account than those of nearby ones. The paper proposes an innovative and rational way to overcome the oversquashing bias. It also provides compelling theoretical evidence for the proposed approach's merit. The numerical results are promising, though not sufficiently thorough to establish this as being close to state-of-the-art. Nevertheless, the reviewers agree that the work is valuable to the community and should be published.

**Award:**

No

---

### Decision · Program_Chairs · 2022-09-14

Accept